# Effect of Topically Applied *Wikstroemia dolichantha* Diels on the Development of Atopic Dermatitis-Like Skin Symptoms in Mice

**DOI:** 10.3390/nu11040914

**Published:** 2019-04-23

**Authors:** Jonghwan Jegal, No-June Park, Tae-Young Kim, Sangho Choi, Sang Woo Lee, Jin Hang, Su-Nam Kim, Min Hye Yang

**Affiliations:** 1College of Pharmacy, Pusan National University, Busan 46241, South Korea; jhjegal@pusan.ac.kr (J.J.); taeyour@pusan.ac.kr (T.-Y.K.); 2Natural Products Research Institute, Korea Institute of Science and Technology, Gangneung 25451, South Korea; 115519@kist.re.kr; 3International Biological Material Research Center, Korea Research Institute of Bioscience and Biotechnology, Daejeon 34141, South Korea; decoy0@kribb.re.kr (S.C.); ethnolee@kribb.re.kr (S.W.L.); 4Institute of Medicinal Plants, Yunnan Academy of Agricultural Sciences, Kunming 650205, China; jinhang1516@sina.com

**Keywords:** *Wikstroemia dolichantha*, atopic dermatitis, oxazolone, 2,4-dinitrochlorobenzene, skin barrier function

## Abstract

Plants of the genus *Wikstroemia* are traditionally used to treat inflammatory diseases like bronchitis and rheumatoid arthritis. In the present study, the anti-atopic effects of an EtOH extract of *Wikstroemia dolichantha* (WDE) on oxazolone- and DNCB (2,4-dinitrochlorobenzene)-induced dermatitis in mice were investigated. Both ears of BALB/c mice were exposed to oxazolone, and dorsal skins of SKH-1 hairless mice were sensitized with DNCB to induce acute eczematous atopic skin lesions. 1% WDE was applied daily to oxazolone- and DNCB-induced AD mice for two or three weeks, respectively. Total IL-4 and IgE concentrations in serum, transepidermal water loss (TEWL) and skin hydration were assessed. High-performance liquid chromatography/mass spectrometry (HPLC/MS) was used to determine the composition of WDE. Dermal application of 1% WDE grossly and histopathologically improved oxazolone- and DNCB-induced AD skin symptoms. Epidermal thickness and mast cell infiltration were significantly lower in animals treated with WDE than in vehicle controls. Furthermore, in addition to reducing DNCB-induced increases in serum IL-4 (interleukin 4) and IgE (immunoglobulin E) levels, WDE also decreased TEWL and increased skin hydration (indicative of improved skin barrier function). The four flavonoids taxifolin, aromadendrin, padmatin and chamaejasmine were tentatively identified in WDE by HPLC-DAD/QTOF-MS. The above results show WDE protected against oxazolone- and DNCB-induced AD in mice by down-regulating the T_H_2-associated cytokine IL-4 and improving skin barrier function and suggest WDE might be useful for the management of atopic dermatitis.

## 1. Introduction

Atopic dermatitis (AD) is a chronic inflammatory skin disease with a multifactorial pathogenesis, and both genetic and environmental factors determine the expression of the disease [1]. Typical symptoms of AD, like pruritus, scaly skin, lichenification, and serous exudate, usually appear on the face and flexors symmetrically [2]. The lifetime prevalence of AD has been reported to be up to 20% in children, and AD develops before the age of 5 years in 85% of all patients [2]. AD poses a substantial public health burden owing to its high prevalence, considerable morbidity, and cost of care [3,4]. AD may be categorized as extrinsic or intrinsic in accord with serum IgE (immunoglobulin E) levels and the presence or absence of allergen-specific IgE [5]. Interleukin-4 (IL-4) is an immunoregulatory lymphokine produced by T-cells and is considered a key factor of extrinsic AD. In fact, elevated serum IL-4 levels and of IL-4 receptor mRNA expression in affected skin are characteristic of extrinsic AD [6,7]. The application of moisturizers is widely recommended as a first-line treatment for AD and tpical corticosteroid and calcineurin inhibitors, such as tacrolimus and pimecrolimus, are used to relieve the symptoms of AD as second-line therapies [8].

Polyphenols are widely distributed in plants and flavonoids are polyphenols, which are polyphenolic secondary metabolites synthesized by numerous plants, and are consumed in significant amounts [9]. Flavonoids share a common 3-ring phenyl benzopyrone structure and are present in nature as free aglycones or glycosides [10]. Flavonoids can be classified as anthocyanins, flavans, flavones, flavanols, flavonols, flavanones, flavonones, isoflavones or others based on the degree of oxidation of the middle pyrone ring or substitution patterns [10,11]. Flavonoids have a broad spectrum of biological activities that include anti-inflammatory, anti-thrombotic, anti-cancer, vasodilating, anti-hepatotoxic, and anti-osteoporotic effects [11,12]. Recently, studies found the anti-inflammatory and anti-allergic properties of flavonoids might be useful for the treatment of AD [13,14,15].

The genus *Wikstroemia*, commonly referred to as ‘Akia’, belongs to the Thymelaeaceae family and is widely distributed in northern Asia through the Himalayas, Malaysia, Oceania, and Polynesia to the Hawaiian Islands [16]. In China, *Wikstroemia indica*, called ‘liao ge wang’ by the Chinese, has been used to treat syphilis, arthritis, pertussis, and cancer [17]. *Wikstroemia* species contain bioactive compounds, which include flavonoids, coumarins and lignans [18,19] and *Wikstroemia* extracts have been reported to possess anti-inflammatory, anti-fungal, and anti-viral properties in in vitro experiments [20,21]. Although several studies have been performed on diverse species of *Wikstroemia* genus, comparatively few have been conducted on *Wikstroemia dolichantha*, and no attempt has been made to investigate its effect on skin barrier function and anti-inflammatory properties. The present study was undertaken to investigate the anti-atopic activities of *W. dolichantha* extract (WDE) in oxazolone or DNCB (2,4-dinitrochlorobenzene)-induced murine models of AD and to document phytochemicals present in the plant.

## 2. Materials and Methods

### 2.1. Plant Material and Extraction

The aerial parts of *Wikstroemia dolichanta* Diels were collected in Yunnan province, Lijiang area, China, and identified by Dr. Sang Woo Lee (Korea Research Institute of Bioscience and Biotechnology). A voucher specimen (PNU-0024) was deposited at the Medicinal Herb Garden, Pusan National University. Dried aerial parts of *W. dolichanta* (5 kg) were extracted with 95% EtOH and evaporated under reduced pressure and freeze-dried to yield *W. dolichanta* EtOH extract (WDE) (466.8 g, 9.3% extract yield).

### 2.2. Animals

Six-week-old female BALB/c and SKH-1 hairless mice were purchased from Orient Bio Inc. (Seongnam, Republic of Korea). Mice were maintained under controlled conditions (25 ± 5℃ and 55 ± 5% RH) with continuous ventilation and fed water and laboratory standard food *ad libitum*. All experimental processes were carried out in accord with the Guide for the Care and Use of Laboratory Animals published by the National Institute of Health (NIH publication no. 85-23, revised 1996) and approved by the Institutional Animal Care and Use Committee of KIST (Certification no. KIST-2016-011).

### 2.3. Measurements of Skin Inflammation in The Oxazolone-Induced Model and WDE Treatment

AD was induced on the ears of BALB/c mice by applying 20 μL of oxazolone (4-ethoxymethylene-2-phenyl-2-oxazolin-5-one) in vehicle (propylene glycol: EtOH = 7: 3) as previously described [22]. One week after first challenge, 20 μL of 0.1% oxazolone was applied every other day to both ears for 3 weeks (the OX group). On the other hand, the ears of BALB/c mice were exposed to 20 μL of 1% WDE twice daily over the same 3-week period (the OX-WDE group). WDE was applied 4 h before and after oxazolone application. The normal control group (CON group) was treated with distilled water only. On the last experimental day, no materials were applied and inflammatory indices, e.g., ear swelling and erythema, were obtained.

### 2.4. Measurement of Skin Severity in The DNCB-Induced Model and Treatment with WDE

DNCB (100 μL) (Sigma-Aldrich, Seoul) was used to provoke AD in SKH-1 hairless mice. During the first 7 experimental days, 1% DNCB was rubbed into the dorsal skin of hairless mice every day. Subsequently, mice were challenged with 0.1% DNCB (100 μL) every 3 days for an additional 2 weeks (DNCB group). In addition, 100 μL of 1% WDE (DNCB-WDE group) or 1% Elidel cream (DNCB-Elidel group) as positive control was applied to the dorsal skins of hairless twice daily over this 2-week period. When DNCB and WDE were applied on same days, WDE was applied 4 h before and after applying DNCB. Normal controls (the CON group) were administered distilled water.

### 2.5. Histological Examinations

To assess histopathological variations, ear skins from BALB/c mice or dorsal skins from SKH-1 hairless mice were fixed in 10% formalin for 24 h and paraffin infiltrated. Tissue sections were sliced to 2℃3 mm, placed on slides, dried overnight at 37℃, and stained with H&E (hematoxylin and eosin) or toluidine blue. Histopathological examinations were performed and photographs obtained using an optical microscope (Olympus CX31/BX51, Olympus Optical Co., Tokyo) fitted with a TE-2000U camera (Nikon Instruments Inc. Melville, USA).

### 2.6. Transepidermal Water Loss (TEWL) and Skin Hydration

TEWL, skin hydration and skin pH values of SKH-1 hairless mice were measured to assess skin barrier status using a Tewameter TM210 (Courage and Khazaka, Cologne, Germany) and a SKIN-O-MAT (Cosmomed, Ruhr, Germany). These three parameters were measured weekly under controlled conditions (25 ± 5 °C, 55 ± 5% RH).

### 2.7. Total Serum IgE and IL-4 Levels

Blood specimens obtained from the abdominal aorta of SKH-1 hairless mice were centrifuged at 10,000 rpm for 15 min at 4℃. Serum samples obtained on experimental day 21 were stored at −70 °C until required for IgE and IL-4 determinations. Total serum IgE and IL-4 levels were measured using ELISA (enzyme-linked immunosorbent assay) kits (eBioscience, San Diego, USA).

### 2.8. HPLC/MS Analysis of WDE

This analysis was performed using an Agilent 6530 Accurate-Mass Q-TOF LC/MS system (Agilent Technologies). The analysis was performed using a Poroshell 120 EC-C18 column (3.0 × 100 mm, 2.7 μm, Agilent) at a flow rate of 0.3 mL/min, and detection was performed at 254 nm. The mobile phase consisted of acetonitrile (A) and ultrapure water (B) and the flow rate was 0.3 mL/min. The gradient profile was as follows: 0–5 min 10% A; 5–30 min, linear gradient from 10 to 70% A; and 30–35 min 70% A. All acquisitions were performed in positive ionization mode. Mass spectra were recorded over the range *m/z*=100–1500 and accurate mass measurements were obtained for all peaks.

### 2.9. Statistical Analysis

The analysis was performed using ANOVA (one-way analysis of variance) and a statistical software program. Results are presented as means ± SEMs and significance was accepted for *p* values < 0.05.

## 3. Results

### 3.1. Effects of WDE on Oxazolone-Induced AD Mice

The experimental procedure is summarized in Figure 1A. When BAKB/c mice were exposed to oxazolone (OX) for 3 weeks, ears exhibited severe AD-like skin symptoms (Figure 1B). However, when mice were cotreated with oxazolone and 1% WDE, these symptoms were reduced. On experimental day 28, 1% WDE co-treatment was found to have markedly prevented lesion formation as compared with OX-treated mice. In fact, erythema, erosion, and dryness, were significant lower in the OX-WDE group than in the OX group on experimental day 28.

### 3.2. Histopathological Analysis of Oxazolone-Treated Skins

Histological changes in the dorsal skins of oxazolone-sensitized BALB/c mice were observed by H&E and toluidine blue staining. As shown in Figure 2A, H&E staining showed dorsal skin sections were thicker in the OX group than in the CON group. Toluidine blue staining also confirmed an increased infiltration of inflammatory cells in the oxazolone-induced CON group (Figure 2B). Ear and epidermal thicknesses and mast cell infiltration counts are shown in Figure 2C–E, respectively. Epidermal thicknesses and numbers of mast cells were significantly lower in the OX-WDE group (40.7 ± 5.8% decrease and 48.9 ± 4.6% decrease, respectively) than in the OX group.

### 3.3. Effects of WDE on Skin Severity in The DNCB-Induced Model and Histopathological Analysis

The experimental procedure is summarized in Figure 3A. Topical application of 1% WDE for two weeks to SKH-1 hairless mice exposed to DNCB effectively reduced AD symptoms including erythema, erosion and hemorrhage (Figure 3B). Dorsal skins collected on experimental day 21 were subjected to H&E staining to determine epidermal thickness or toluidine blue staining to detect cells infiltrating tissues. Epidermal thickening and inflammatory cell infiltration were observed in the DNCB group, although they were not as high as the DNCB-WDE group (Figure 4A,B). WDE reduced DNCB-induced epidermal thickness and mast cell numbers by 57.1 ± 6.2% and 16.7 ± 2.3%, respectively, as compared with the DNCB group (Figure 4C,D). Application of 1% pimecrolimus cream (Elidel) as a positive control improved AD skin symptoms by decreasing epidermal thickness (Figure 4A,C) and mast cell infiltration (Figure 4B,D) in DNCB-induced atopic animal model.

### 3.4. Effects of WDE on Blood IL-4 and IgE Levels in The DNCB Model

We investigated the effects of WDE on the expressions of inflammation-related factors, that is, IL-4 and IgE, in the sera of DNCB-stimulated SKH-1 mice. As shown in Figure 5, DNCB strongly increased serum IgE and IL-4 concentrations, but 1% WDE co-treatment noticeably decreased total IgE (CON: 100.4 ng/mL, DNCB: 465.3 ng/mL and DNCB-WDE: 428.9 ng/mL) (Figure 5A) and IL-4 (CON: 16.9 pg/mL, DNCB: 43.9 pg/mL and DNCB-WDE: 37.6 pg/mL) levels (Figure 5B). 1% Elidel co-treatment inhibited DNCB-induced IgE increase (CON: 100.4 ng/mL, DNCB: 465.3 ng/mL and DNCB-Elidel: 308.3 ng/mL) (Figure 5A) and IL-4 increase (CON: 16.9 pg/mL, DNCB: 43.9 pg/mL and DNCB-Elidel: 35.9 pg/mL) (Figure 5B).

### 3.5. Effects of WDE on Skin Barrier Function in DNCB-Treated SKH-1 mice

Skin barrier functions were evaluated by measuring TEWL and skin hydration. DNCB treatment decreased stratum corneum water content, but two weeks of 1% WDE enhanced skin hydration as compared with the DNCB group (Figure 5). Mean TEWL was 12.5 ± 7.5% lower in the DNCB-WDE group than in the CON group (Figure 5C) and skin hydration was significantly greater in the DNCB-WDE group than in the CON group (Figure 5D). On day 21, co-treatment with 1% Elidel, a positive control, reduced the decrease in TEWL observed in the DNCB control by 28.5 ± 5.2% (Figure 5C) and increased skin hydration by 12.0 ± 8.2% (Figure 5D).

### 3.6. HPLC/MS of WDE

Optimization of the mobile phase to acetonitrile/water (1:9→70% acetonitrile, gradient system) resulted in a satisfactory S/N at a wavelength of 254 nm for all detected peaks. The presence of four compounds, that is, 1: taxifolin (*m/z* 303.05 at *t*_R_ 14.4 min), 2: aromadendrin (*m/z* 287.06 at *t*_R_ 18.3 min), 3: padmatin (*m/z* 317.07 at *t*_R_ 21.2 min) and 4: chamaejasmine (*m/z* 539.24 at *t*_R_ 23.3 min), was confirmed by comparing retention times and UV spectra with standard compounds (Figure 6).

## 4. Discussion

Plant extracts have been adopted as complementary and alternative medicines for the treatment and/or prevention of mild-to-severe AD [23]. Somewhat surprisingly, the major causes of skin diseases like atopic dermatitis (AD), psoriasis and pruritus have not yet been clearly identified and currently, the pathogenesis of AD is believed to involve complex interrelationships between genetic, environmental, skin barrier, pharmacologic, psychologic and immunologic factors [23,24,25]. Recent studies have indicated that skin barrier disruption contributes to the onset of AD [25,26]. Skin barrier disruption is the hallmark of AD and represents the main cause of the subsequent release of pro-inflammatory mediators [27]. Certain emollients or moisturizers, such as ceramide and hyaluronic acid, enhance skin barrier function, and thus, it has been suggested that barrier-strengthening moisturizers might delay the relapse of AD [28]. Topical corticosteroids are used in the management of AD as a second-line therapy and long-term treatment is problematic due to steroid-induced side effects, such as, skin atrophy and telangiectasia [8]. Accordingly, efforts are being made to identify naturally derived agents because they are likely to provide safer, more effective therapies.

In our preliminary in vitro experiment, we found that WDE had inhibited IL-4 production in RBL-2H3 cells (data not shown). Based on this result, we conducted a series of in vivo experiments to investigate its potential as a therapeutic agent for AD. The ears of female BALB/c mice were sensitized with 1% oxazolone for 1 week and then treated topically with 0.1% oxazolone for 3 weeks to induce AD-like skin symptoms. After oxazolone treatment mice showed typical skin lesions of AD (dry skin, cutaneous thickening, erythema, and edema). However, when 1% WDE was applied during the same 3-week period, these symptoms were significantly alleviated, as were oxazolone-induced epidermal thickening and mast cell infiltration. Topical treatment with WDE was found to have an anti-AD effect in our oxazolone-induced murine AD model.

To determine whether the topic application of WDE relieves atopy, additional in vivo testing was conducted using a DNCB-induced SKH-1 hairless mouse model of AD. The dorsal skins of hairless mice were treated with 1% DNCB daily for 1 week and then with 0.1% DNCB every 3 days for 2 weeks. The dermal irritation induced by this treatment provoked the expression of IL-4 and massive release of IgE to serum. AD, associated with T_H_1 type cytokine IFN-γ and T_H_2 response, increased levels of T_H_2-associated cytokines (IL-4, IL-5 and IL-13) [29]. Herbal anti-inflammatory agents are often used to prevent and treat skin and allergic diseases [30]. Plant-derived phytochemicals with anti-AD properties are known to exert their effects by disrupting the activities of inflammatory cytokines and receptors [30,31]. In the present study, treatment with 1% WDE during the final 2 weeks of DNCB application significantly reduced DNCB-induced increases in serum IL-4 (the critical T_H_2 cytokine) and IgE production. Furthermore, the lesioned skins of DNCB-treated mice had greater TEWL and lower skin hydration values than treatment naïve controls, indicating impaired barrier function [32,33]. In particular, the 1% WDE treatment produced the similar inhibitory effect when compared to the positive control, 1% Elidel, regarding reducing serum IL-4 level. Application of WDE during the last 2 weeks of DNCB application markedly improved skin biophysical properties of dry atopic skin versus DNCB application alone. These results indicate WDE acts as a moisturizer and improves skin barrier function in atopic skin and reduces atopic response to irritants.

Nine flavonoids, that is, apigenin, afzelechin, afzelechin 3-*O*-glucoside, aromadendrin, catechin, kaempferol, taxifolin, padmatin, and chamaejasmine, were identified in the aerial parts of *W. dolichantha* during the present study (unpublished data). Furthermore, phytochemical screening of WDE revealed the presence of the flavonoids taxifolin, aromadendrin, padmatin, and chamaejasmine. Flavonoids are phenolics commonly found in fruits, vegetables, grain, flowers, tea, and wine [34] that possess diverse bioactivities [11]. However, the role of flavonoids in human health have been inconclusive and their potential use, especially in infants and children, is still a matter of debate [35]. Taken together, these four flavonoids would appear to be responsible for the anti-atopic effects of WDE.

## 5. Conclusions

In summary, an EtOH extract of the aerial parts of *W. dolichantha* (WDE) was found to effectively reduce the AD-like symptoms induced by oxazolone or DNCB via regulating IgE synthesis and T_H_2-mediated cytokine IL-4 expression. Topical treatment with WDE strengthens the skin barrier function, thereby exhibiting an excellent effect of improving skin moisturization in our BALB/c and SKH-1 mouse models with AD. Phytochemical analysis supports the hypothesis that the anti-inflammatory and anti-allergic effects of WDE could, in part, be due to the presence of flavonoids like taxifolin, aromadendrin, padmatin, and chamaejasmine as major constituents. Our results suggest that *W. dolichantha* may offer a potential means of preventing/treating atopy.

## Figures and Tables

**Figure 1 nutrients-11-00914-f001:**
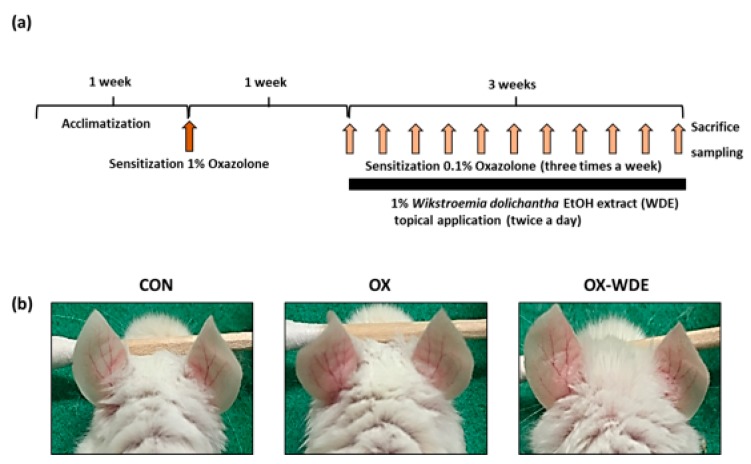
Effects of WDE on the development of oxazolone-induced AD-like symptoms in the ears of BALB/c mice. (**a**) Schematic representation of the experiment. (**b**) Clinical features of AD-like skin lesions. WDE: EtOH extract of *Wikstroemia dolichantha*, CON: control group, OX: oxazolone-treated group, OX-WDE: oxazolone and 1% *W. dolichantha* EtOH extract-treated group.

**Figure 2 nutrients-11-00914-f002:**
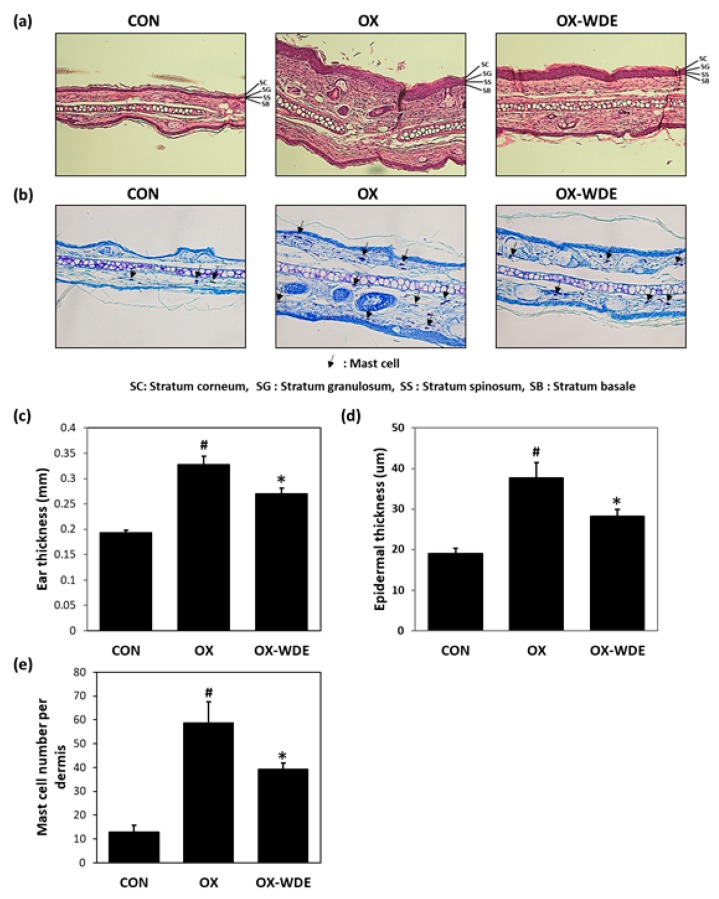
Effects of WDE on histopathological changes, ear thicknesses, epidermal thicknesses, and mast cell numbers on oxazolone-induced AD-like symptoms in BALB/c mouse ears. (**a**) Hematoxylin and eosin (H&E) staining. (**b**) Toluidine blue staining. (**c**) Ear thicknesses. (**d**) Epidermal thicknesses. (**e**) Mast cell numbers. Results are expressed as the means ± SEMs (*n* = 7) of two independent experiments performed in triplicate. ^#^*p* < 0.05 vs. the CON group; ^*^*p* < 0.05 vs. the OX group.

**Figure 3 nutrients-11-00914-f003:**
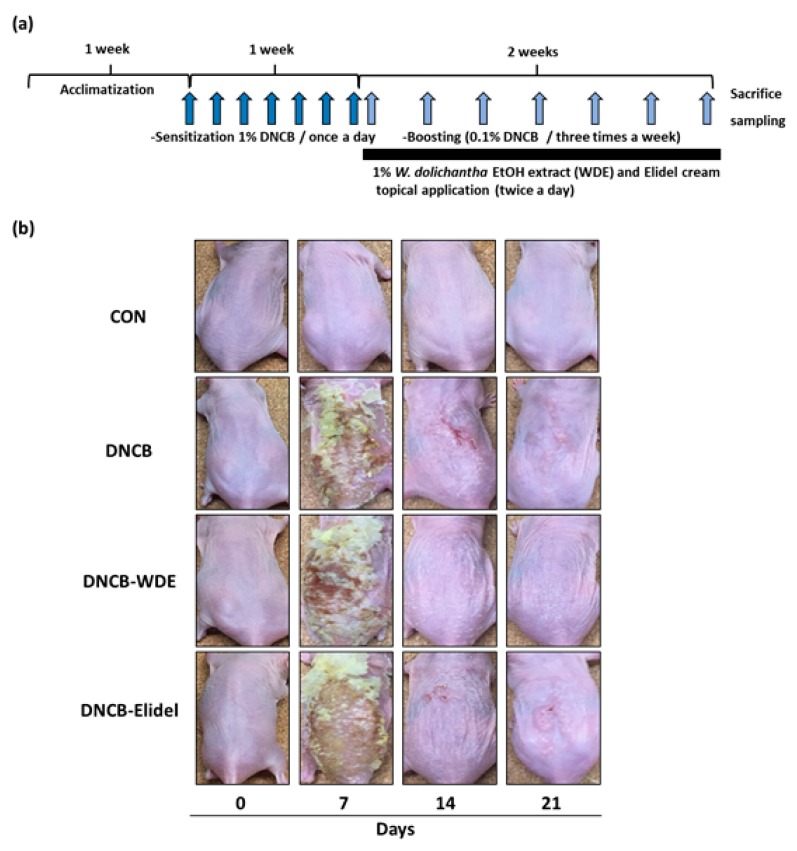
Effects of WDE on the development of DNCB-induced AD-like symptoms in hairless mice. (**a**) Schematic representation of the experiment. (**b**) Clinical features of AD-like skin lesions. WDE: EtOH extract of *Wikstroemia dolichantha*, CON: vehicle control group, DNCB: DNCB-treated group, DNCB-WDE: DNCB and 1% WDE-treated group, DNCB-Elidel: DNCB and 1% Elidel-treated group.

**Figure 4 nutrients-11-00914-f004:**
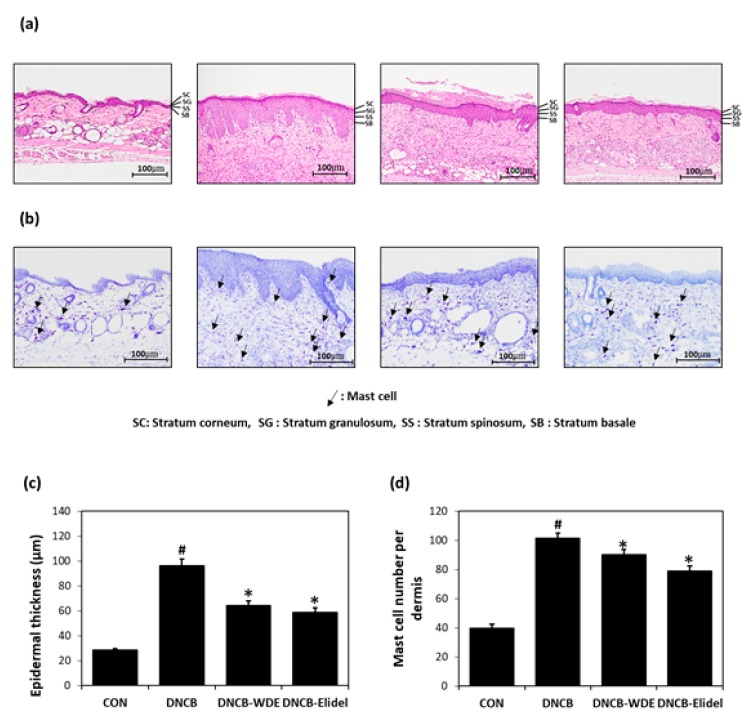
Effects of WDE on histopathological changes, epidermal thicknesses, and mast cell numbers on DNCB-induced AD-like symptoms in SKH-1 hairless mice. (**a**) H&E staining. (**b**) Toluidine blue staining. (**c**) Epidermal thicknesses. (**d**) Mast cell numbers. Results are expressed as the means ± SEMs of two independent experiments. ^#^*p* < 0.05 vs. the CON group; * *p* < 0.05 vs. the DNCB group. WDE: EtOH extract of *Wikstroemia dolichantha*, CON: control group, DNCB: DNCB-treated group, DNCB-WDE: DNCB and 1% WDE-treated group, DNCB-Elidel: DNCB and 1% Elidel-treated group.

**Figure 5 nutrients-11-00914-f005:**
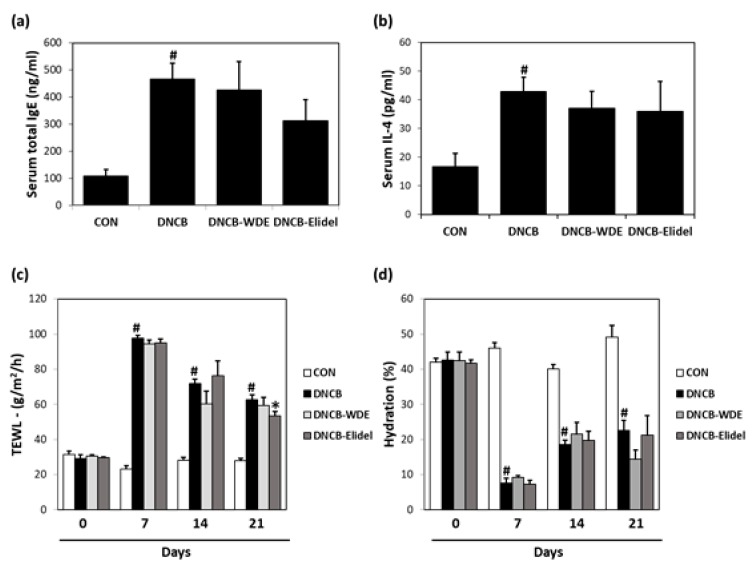
Effects of WDE on skin barrier function, serum IgE, and serum IL-4 on DNCB-induced AD-like symptoms in SKH-1 hairless mice. (**a**) Serum total IgE levels. (**b**) Serum total IL-4 levels. (**c**) Transepidermal water loss (TEWL). (**d**) Skin hydration values. Results are expressed as the means ± SEMs (*n* = 7) of two independent experiments. ^#^*p* < 0.05 vs. the CON group; * *p* < 0.05 vs. the DNCB group. EtOH extract of *Wikstroemia dolichantha*, CON: control group, DNCB: DNCB-treated group, DNCB-WDE: DNCB and 1% WDE-treated group, DNCB-Elidel: DNCB and 1% Elidel-treated group.

**Figure 6 nutrients-11-00914-f006:**
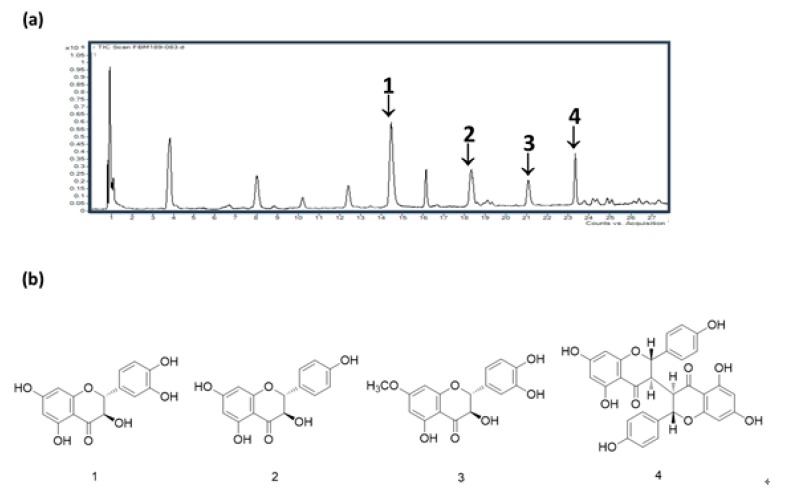
(**a**) HPLC/MS chromatogram of WDE. (**b**) Chemical structures of its major components. Fingerprint analysis of WDE was performed in positive ion mode by HPLC/MS. 1: taxifolin, 2: aromadendrin, 3: padmatin, 4: chamaejasmine.

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
