# Peer review of "Effect of Topically Applied Wikstroemia dolichantha Diels on the Development of Atopic Dermatitis-Like Skin Symptoms in Mice"

_nutrients, 2019, doi:10.3390/nu11040914_

Reviewer 1 Report

Plants of the genus Wikstroemia are traditionally used to treat inflammatory diseases, such as, bronchitis and rheumatoid arthritis : are traditionally used to treat inflammatory diseases like bronchitis and rheumatoid arthritis.

To investigate the anti-atopic effects of an EtOH extract of Wikstroemia dolichantha (WDE) on oxazolone-induced atopic dermatitis (AD)-like skin lesions in the ears of oxazolone sensitized BALB/c mice and dorsal skins of DNCB (2,4- dinitrochlorobenzene) sensitized AD hairless mice : Too long sentences, confused and without verbs.

3.2. Histopathological Analysis of Oxazolone-Treated Skins Histological changes in the dorsal skins of oxazolone-sensitized BALB/c mice were observed by H&E and toluidine blue staining. As shown in Fig 2A, H&E staining showed dorsal skin sections were thicker in the OX group than in the CON group. Toluidine blue staining also confirmed an increased infiltration of inflammatory cells in the oxazolone-induced CON group (Fig 2B). Ear and epidermal thicknesses and mast cell infiltration counts are shown in Figs 2C, 2D, and 2E, respectively. Epidermal thicknesses and numbers of mast cells were significantly lower in the OXWDE group (40.7% decrease and 48.9% decrease, respectively) than in the OX group.

 Where is standard deviation of  red underlined values?

3.3, 3.4,3.5: I think it is better to add standard deviation of percentage values reported.

Conclusions: I suggest to restructure and to stress the scientific soundness of the work. I think the conclusions presented in this way are only a repetition of discussion paragraph.

Finally paper needs an extensive editing of English language and style required

Author Response

Reviewer #1

Plants of the genus Wikstroemia are traditionally used to treat inflammatory diseases, such as, bronchitis and rheumatoid arthritis : are traditionally used to treat inflammatory diseases like bronchitis and rheumatoid arthritis.

- We appreciate reviewer’s valuable comment and now we revised it as suggestion.

To investigate the anti-atopic effects of an EtOH extract of Wikstroemia dolichantha (WDE) on oxazolone-induced atopic dermatitis (AD)-like skin lesions in the ears of oxazolone sensitized BALB/c mice and dorsal skins of DNCB (2,4- dinitrochlorobenzene) sensitized AD hairless mice : Too long sentences, confused and without verbs.

- It is now changed to ‘. In the present study, the anti-atopic effects of an EtOH extract of Wikstroemia dolichantha (WDE) on oxazolone- and DNCB (2,4-dinitrochlorobenzene)-induced dermatitis in mice were investigated.’ (Please see yellow-highlighted part).

3.2. Histopathological Analysis of Oxazolone-Treated Skins Histological changes in the dorsal skins of oxazolone-sensitized BALB/c mice were observed by H&E and toluidine blue staining. As shown in Fig 2A, H&E staining showed dorsal skin sections were thicker in the OX group than in the CON group. Toluidine blue staining also confirmed an increased infiltration of inflammatory cells in the oxazolone-induced CON group (Fig 2B). Ear and epidermal thicknesses and mast cell infiltration counts are shown in Figs 2C, 2D, and 2E, respectively. Epidermal thicknesses and numbers of mast cells were significantly lower in the OXWDE group (40.7% decrease and 48.9% decrease, respectively) than in the OX group.

 Where is standard deviation of red underlined values?

- Standard deviations are now newly inserted like 40.7 ± 5.8% decrease and 48.9 ± 4.6% decrease’.

3.3, 3.4,3.5: I think it is better to add standard deviation of percentage values reported.

- Standard deviations are now newly inserted like 57.1 ± 6.2% and 16.7 ± 2.3%’ for 3.3. and ‘12.5 ± 7.5%, 28.5 ± 5.2% (Fig 5C) and increased skin hydration by 12.0 ± 8.2%’ for 3.5. (Please see yellow-highlighted parts).

Conclusions: I suggest to restructure and to stress the scientific soundness of the work. I think the conclusions presented in this way are only a repetition of discussion paragraph.

- As suggested by reviewer, we rewrote the conclusion. Please see yellow-highlighted parts in the revised version of manuscript.

Finally paper needs an extensive editing of English language and style required

- We had English editing service before manuscript submission.

Author Response

Reviewer #2

The objective of the study is interesting and the experimental sections are well detailed. However, some issues should be explained, as following:

In the Introduction section the authors state that “AD chronic inflammatory skin disease associated with cutaneous overactivity to factors that do not affect healthy individuals  [1]”. I fell that this statement is partly due, as AD is more complex as it represents a multifactorial and heterogenous disease, including different clinical phenotypes, due to the interplay of genetic and multiple environmental factors, and “cutaneous overactivity” is only one of these (I suggest to see and cite the most recent review by Weidinger S, Novak N. Atopic dermatitis. Lancet. 2016; 387:1109- 22). Accordingly, I suggest to moderately rewrite the sentence

A. We rewrote the sentence as suggested by reviewer like ‘Atopic dermatitis (AD) is a chronic inflammatory skin disease with a multifactorial pathogenesis and both genetic and environmental factors determine the expression of the disease [1].’ and the Ref. 1 is now changed to ‘Weidinger S, Novak N. Atopic dermatitis. Lancet. 2016; 387:1109- 22’. Please see yellow-highlighted part.

Introduction section: … AD is associated with severe itching, and thus, decreases mental health. related quality of life. I think the authors mean the global burden of AD on patient quality of life (not only the mental-health related quality of life). I suggest to better explicit this concept (see and cite  also Carroll, C.L.; Balkrishnan, R.; Feldman, S.R.; Fleischer, A.B., Jr.; Manuel, J.C. The Burden of Atopic Dermatitis: Impact on the Patient, Family, and Society. Pediatr. Dermatol. 2005 and Boccardi, D.; D’Auria, E.; Turati, F.; DI Vito, M.; Sortino, S.; Riva, E.; Cerri, A. Disease Severity and Quality of Life in Children with Atopic Dermatitis: PO-SCORAD in Clinical Practice. Minerva Pediatr. 2017, 69, 373–380)

A. The sentence is now corrected to ‘AD poses a substantial public health burden owing to its high prevalence, considerable morbidity, and cost of care [3,4].’ and we cited two references as suggested by reviewer. Please see yellow-highlighted part.

Introduction section: “Topical corticosteroid and calcineurin inhibitors, such as, tacrolimus and pimecrolimus, are mainly used to relieve the symptoms of AD”[3]. In this statement it lacks the important role of moisturizers that as first-line recommended treatments; topical steroids and calcineurin inhibitors should be used as second-line treatments for treating AD flares (see and cite D’Auria, E.; Banderali, G.; Barberi, S.; Gualandri, L.; Pietra, B.; Riva, E.; Cerri, A. Atopic Dermatitis: Recent Insight on Pathogenesis and Novel Therapeutic Target. Asian Pac. J. Allergy Immunol. 2016)

A. We now changed it to ‘The application of moisturizers is widely recommended as a first-line treatment for AD and tpical corticosteroid and calcineurin inhibitors, such as, tacrolimus and pimecrolimus, are used to relieve the symptoms of AD as second-line therapies [7].’ to emphasize the importance of moisturizers as first-line therapy of AD. Previous reference is also changed to ‘D’Auria, E.; Banderali, G.; Barberi, S.; Gualandri, L.; Pietra, B.; Riva, E.; Cerri, A. Atopic Dermatitis: Recent Insight on Pathogenesis and Novel Therapeutic Target. Asian Pac. J. Allergy Immunol. 2016.’

Introduction section:.. Wikstroemia extracts have been reported to possess anti-inflammatory, anti-fungal and anti-viral properties: where these effects have been observed? In vitro experiments, animal models or other? The authors should clarify

 A. To clarify the sentence, it is changed to ‘Wikstroemia extracts have been reported to possess anti-inflammatory, anti-fungal, and anti-viral properties in in vitro experiments’.

 Introduction section: …to investigate its anti-atopic effects. As it refers to the effects observed, I believe it should be more correct to use “to investigate its effect on skin barrier function and antinflammatory properties” instead of “anti-atopic effects” that may be misleading in the context

 A. We agree with reviewer’s opinion and the sentence is now corrected to ‘to investigate its effect on skin barrier function and anti-inflammatory properties’

 Materials and methods: aerial parts of W. dolichanta (5 kg) were extracted with 95% EtOH

A curiosity: how much is the ethanol concentration after eliminating the extract solvent?

A. We achieved the final extract after freeze drying of sample, indicating that nearly complete removal of the solvent. Sample preparation method is described as ‘Dried aerial parts of W. dolichanta (5 kg) were extracted with 95% EtOH and evaporated under reduced pressure and freeze-dried to yield W. dolichanta EtOH extract (WDE) (466.8 g, 9.3% extract yield).’. Please see yellow-highlighted part.

Discussion: …. skin barrier disruption contributes to the onset of AD. Skin barrier disruption is the hallmark of AD and represents the main cause of the subsequent release of pro-inflammatory mediators (also see and cite Lee SH. Epidermal permeability barrier defects and barrier repair therapy in atopic dermatitis. Allergy Asthma Immunol Res. 2014;6:276-87)

 A. We newly added that the sentence with the reference. Please see yellow-highlighted part.

 Discussion:. “Topical corticosteroids are usually used as a first line therapy for AD”-  This statement is not correct: antinflammatory drugs, such as steroids, are a second-line therapy, used to treat reacutization. (see Eichenfield LF, Tom WL, Berger TG, Krol A, Paller AS, Schwarzenberger K, et al. Guidelines of care for the management of atopic dermatitis: section 2. Management and treatment of atopic dermatitis with topical therapies. J Am Acad Dermatol. 2014;71:116-132)

 A. We agree with reviewer’s opinion and the sentence is now changed to ‘Topical corticosteroids are used in the management of AD as a second-line therapy and long-term treatment is problematic due to steroidinduced side effects, such as, skin atrophy and telangiectasia [7].’ Please see yellow-highlighted part.

 AD is mainly associated with TH2 response: this statement is not completely true as the role the role of Th1 and Th17 has been well demonstrated in AD and AD is often “non atopic”

 A. We agree with reviewer’s opinion and the sentence is now corrected to ‘’AD is associated with TH1 type cytokine IFN-γ and TH2 response, increased levels of TH2-associated cytokines (IL-4, IL-5 and IL-13) [27].’

The role of flavones in human nutrition and their potential adverse effects on human health have been discussed in the past years; therefore, their potential use, especially in infants and children, is still a matter of debate. Considering that AD is more prevalent in the first years of life (about 60%) can the authors comment about this concern with a brief comment in the discussion section?

A. It is now newly inserted like ‘However, the role of flavonoids in human health have been inconclusive and their potential use, especially in infants and children, is still a matter of debate [33].’ In Discussion part.

 Conclusions: considering the study nedds further confimration, it should be better conclude that it may offer a potential means of preventing/treating atopy

 A. As suggested by review, it is now corrected to ‘W. dolichantha may offer a potential means of preventing/treating atopy’.